

# Frequency analysis of extreme floods in a highly developed river basin

Tomohiro Tanaka[1], Yasuto Tachikawa[1], Yutaka Iachikawa[1], Kazuaki Yorozu[1]

[1]Graduate School of Engineering, Kyoto University

*Correspondence to*: Tomohiro Tanaka (tanaka.tomohiro.65m@kyoto-u.jp)

**Abstract.** Design flood, river discharge with a particular return period, is fundamental to determine the scale of flood control facilities. In addition, considering a changing climate, not only frequencies of river discharge at design level but those of devastating flooding are also crucial. Characteristics of river discharge during extreme floods largely differ from those during others because of upstream dam operation and/or river overflow; however, flood frequency analysis (FFA) from past

discharge data is difficult to represent such impact because river basins rarely experience floods over the design level after river improvement and dam construction. To account for the above impact on extreme flood frequencies, this study presented a rainfall-based flood frequency model (RFFM) that derives flood frequencies from probabilistic rainfall modelling that empirically represents probabilistic structure of rainfall intensity over a catchment by directly using observed spatial-temporal rainfall profiles. The RFFM was applied to the Yodo River basin, Japan and demonstrated that flood

frequency estimations by the RFFM well represent past flood frequencies at the Hirakata gauging station. Furthermore, the RFFM showed that return periods of large flood peaks are estimated at extremely large values, reflecting decrease of discharge by the inundation in an upstream area of the gauging station. On the other hand, FFA from past discharge data did not represent this impact because it has not experienced such huge flood peaks in an observation period. This study demonstrated the importance of the RFFM for flood frequency estimations, including those exceeding the design level.

**1 Introduction**

Design flood determines the size of flood control structures such as dams and/or river dikes and is estimated from the probability distribution of annual maximum flood peak discharge (AMF). For the purpose, flood frequency analysis (FFA) has been mainly aimed at frequency estimations at the design level. In addition, to make adaptation plans against recent devastating flood disasters and climate change, the rational estimation of extreme floods over the design level is also crucial.

Such floods may cause overflow and/or river bank failure, which cause drastic change in downstream discharges. Usually, a river basin of interest has not ever experienced this event; therefore, this impact is not reflected in the probability distribution of AMF estimated from historical discharge data. Flood frequency estimations incorporating the impact of upstream inundation during extreme floods are expected to differ from those without this consideration. Therefore, this study aims to



reasonably estimate return periods of floods exceeding the design level, by accounting for the decrease of downstream discharge by upstream inundation.

There are two major approaches to estimate the probability distribution of AMF: rainfall-runoff simulation of flood peak discharge from design storms (rainfall-based approach) and FFA of historical discharge data (discharge-based approach). The discharge-based approach is straightforward; however, the statistical characteristics of discharge might change over time because of river basin changes such as flood control facilities and/or urban development. Moreover, it is difficult to estimate frequencies of floods over the design level because such floods, which are affected by the upstream dam/inundation, does rarely occur and thus is not usually included in historical discharge data. On the other hand, rainfall-based approaches are capable to consider the impact of anthropogenic changes of rive discharge during floods by generating extreme rainfall events.

In past decades, rainfall-based approaches to flood frequency estimation and/or flood risk assessment have been demonstrated in a number of studies [*Eagleson,* 1972; *Boughton and Droop*, 2003; *Rogger et al.,* 2012; *Candela et al.,* 2014; *England et al.,* 2014], most of which adopted intensity-duration-frequency (IDF) curve [*Chow,* 1953; *Koutsoyiannis et al.,* 1998; *Nhat al.,* 2006] or point process models [*Foufoula-Georgiou,* 1989; *Onof et al.,* 2000; *Wheater et al.,* 2005]. They are mainly applied to small river catchments where a spatially uniform design storm is acceptable or larger river basins by introducing spatial rainfall modeling, which has many parameters to be calibrated with long-term radar data [*Wheater et al.,* 2005]. Another type of rainfall models represents the occurrence of rainfall by Poisson processes [*Eagleson*, 1972] and its temporal profile [*Cameron et al.*, 1999; *Blazkova and Beven*, 2002]. This type of rainfall modelling achieves rational representation of stochastic characteristics of rainfall with smaller computational costs and parameters and has been extended to spatial rainfall modelling by applying a single site model to several sub-catchments [*Blazkova and Beven*, 2004].

On the other hand, in Japan, some large river basins in Japan has applied a simple rainfall-based approach (SRA) , which converts probability distribution of annual maximum basin rainfall into that of AMF accounting for spatial-temporal rainfall variability, for practical design flood determination [*Ohmachi,* 2004]. The SRA describes the occurrence of rainfall events using the Poison process in common with other rainfall process models [*Rodriguez-Iturbe et al.,* 1984; *Acreman,* 1990], with simplified representation of the probabilistic structure of spatial-temporal rainfall patterns by directly using those of past rainfall events [*Shiiba and Tachikawa*, 2013]. Furthermore, *Tanaka et al.* [2015a] addressed its assumption that total basin rainfall and its rainfall pattern are independently determined and proposed a rainfall-based flood frequency model (RFFM), by considering the dependence structure between total basin rainfall and rainfall duration. As the results, the RFFM well represents the frequency distribution of AMF in the Yura-gawa River basin (1,882 km$^2$ ), Japan. Both the RFFM and rainfall modelling by *Cameron et al.* [1999] directly apply temporal profiles of observed rainfall events; however, these spatial modelling of rainfall is different. The RFFM applies the observed rainfall events to both temporal and spatial profiles, while



other rainfall models extend single site models to several points or sub-catchments. More empirical representation of spatial rainfall structure in the RFFM allows design flood estimations in large river basins with simpler rainfall modelling and thus much smaller model parameters. Furthermore, the RFFM derives CDF of AMF without any Monte Carlo simulation; thus, this approach has no issue of simulation periods and random number generation.

This study presents the RFFM and applies it at the Hirakata discharge gauging station in the Yodo River basin (7,280 km$^2$), Japan and compares the estimated probability distribution of AMF with that estimated by FFA of a time series discharge data. Comparison between the RFFM and FFA of observed discharge data is affected by both flood frequency modeling and rainfall-runoff modeling; therefore, to validate flood frequency estimations by the RFFM, FFA in this study is based on a time series data of discharge simulated from historical rainfall events. Through this comparison, we aim to examine how

flood frequency estimations from the RFFM and FFA behave for huge flood peak that has not been observed. Extrapolation by FFA is difficult to represent drastic change in downstream river discharges that are affected by dam operation and/or upstream inundation. This study shows how the RFFM estimates extreme flood frequencies reflecting drastic change of flood frequencies.

Figure 1 shows the framework of this study. The study estimates the probability distribution of AMF by using the RFFM and

discusses the performance of the RFFM, by comparing the results with FFA of discharge data. To discuss the effect of flood control by the upstream dams, the RFFM and FFA are applied by using a rainfall-runoff model 1) without damn operation and an inundation model (WOD-WOI), 2) with dam operation and without an inundation model (WD-WOI), and 3) with dam operation and an inundation model (WD-WI). In section 2, the basic structure of the RFFM is described and then applied to the Hirakata station. Then, in section 3, its application to the Hirakata station in the Yodo River basin is

implemented. In section 5, flood frequency analyses in annual maximum series (AMS) and peak-over-threshold (POT) approaches are performed from simulated discharge data by using rainfall-runoff and/or flood-inundation models; then, flood frequencies estimated by the both approaches, namely the RFFM and FFA, are compared to each other for all the three cases; WOD/WOI, WD/WOI, and WD/WI.

**2 Development of the Rainfall-based Flood Frequency Model (RFFM)**

This section presents the RFFM that derives the probability distribution of AMF from probabilistic rainfall modelling, according to Tanaka et al. [2015a].

**2.1 Probabilistic rainfall modelling**

Over the years, a number of studies demonstrated stochastic rainfall modelling such as the Neyman-Scott model [*Valdes et al.*, 1985], Waymire, Gupta, and Ro-driguez-Iturbe (WGR) model [*Waymine et al.*, 1984; *Rodriguez-Iturbe et al.*, 1987], and



Bartlett-Lewis Rectangular Pulse Model [*Onof and Wheater*, 1993]. These models elaborately state probabilistic structure of a storm event from a rainfall cell level, but, due to multidimensional structure of spatial-temporal distribution of rainfall, they are complicated and not easy to represent extreme rainfall characteristics. Therefore, this study presents much simpler representation of stochastic rainfall modeling based on practical design flood determination on some river basins in Japan [*Ohmachi*, 2004]. This method assumes that the occurrence of rainfall follows the Poison process with similar to the above point process models; however, probabilistic modelling of spatial-temporal rainfall pattern (hereinafter referred to as "rainfall pattern") is largely different from them. The RFFM separates total basin rainfall $r_a$ and its rainfall pattern $\xi$, where the rainfall pattern and rainfall intensity at location $(x, y)$ and time $t$; $\xi(x, y, t)$ and $r(x, y, t)$ satisfy the following relation;

$$r(x, y, t) = r_a \xi(x, y, t), \quad (x, y) \in B, 0 \le t \le d_i \tag{1}$$

where $B$ is the target basin. $d$ is rainfall duration. $\xi_i(x, y, t)$ is normalized to satisfy the following equation;

$$\iint_B \left( \int_0^d \xi(x, y, t) dt \right) dx dy = S \tag{2}$$

where $S$ is the basin area.

The occurrence of a rainfall pattern and total basin rainfall are separately modeled by the following two assumptions;

a) When a rainfall event occurs, one of $N$ rainfall patterns $\xi_i$ $(i = 1, 2, ..., N)$ occurs with the probability $p_i$ and its rainfall duration is $d_i$.

b) Total basin rainfall $r_a$ follows the conditional probability distribution given duration $d_i$, $G_{R_a|D}(r_a | d_i)$

The RFFM simplifies both temporal and spatial profiles of rainfall events by using a large number $N$ of observed rainfall patterns.

## 2.2 Probability distribution of flood peak discharge during a storm event

According to the above two assumptions, the probability with which flood peak discharge $Q_p$ is equal to or less than a particular discharge $Q_{p1}$ is given as follows:

$$\Pr[Q_p \le Q_{p1}] = \sum_{i=1}^N p_i \Pr\left[r_a \le r_{a,i}(Q_{p1}) \mid d = d_i\right] \tag{3}$$





where $r_{a,i}(Q_{p1})$ is basin rainfall that corresponds to flood peak discharge $Q_{p1}$ for the $i$-th rainfall pattern. When fixing rainfall pattern, a storm event with larger total basin rainfall causes larger flood peak discharge (demonstrated in the section 4); thereby, $r_{a,i}(Q_{p1})$ is uniquely determined. The left hand side of equation (3) indicates the non-exceedance probability of flood peak discharge during a storm event $G_{Q_p}(Q_p)$; thus, $G_{Q_p}(Q_p)$ is given as follows:

$$G_{Q_p}(Q_p) = \sum_{i=1}^{N} p_i G_{R_a|D}\left(r_{a,i}(Q_p)\,|\,d_i\right) \qquad (4)$$

Figure 2 illustrates a schematic diagram of equation (4). For a particular discharge $\hat{Q}_p$, its non-exceedance probability is obtained by first obtaining the corresponding basin rainfall for $i$-th rainfall pattern $r_{a,i}(\hat{Q}_p)$; then averaging non-exceedance probability of this rainfall $G_{R_a|D}\left(r_{a,i}(\hat{Q}_p)\,|\,d_i\right)$, weighting by using occurrence probability of the rainfall pattern $p_i$.

**2.3 Procedures to calculate probability distribution of flood peak discharge**

Figure 3 shows the flowchart of the RFFM. As shown in Figure 3, the RFFM calculates $G_{Q_p}(Q_p)$ through the following steps;

1) To prepare rainfall patterns $\xi_i(x, y, t)$ $(i = 1, 2, ..., N)$ from observed $N$ rainfall events.

2) To estimate the conditional CDF of total basin rainfall $r_a$ given rainfall duration $d$, $G_{R_a|D}(r_a\,|\,d)$.

3) To prepare a storm event by extending total basin rainfall of $\xi_i(x, y, t)$ to that with a certain amount $\hat{r}$.

4) To simulate flood peak discharge from the prepared storm event.

5) To increase total basin rainfall $\hat{r}$ and to obtain a relationship between total basin rainfall and flood peak discharge for rainfall pattern $\xi_i(x, y, t)$, $r_{a,i}(Q_p)$ (hereinafter referred to as "R-Q relationships"), by repeating 3) and 4).

6) To repeat steps 3), 4), and 5) for all $N$ rainfall patterns.

7) To obtain non-exceedance probability of flood peak discharge from equation (4) by averaging non-exceedance probability of the corresponding total basin rainfall, $G_{R_a|D}(r_a(Q_p)\,|\,d_i)$ for all $N$ rainfall patterns, weighting it with



occurrence probability of each rainfall pattern $p_i$.

In general, it is difficult to identify occurrence probability of rainfall pattern $p_i$; thus, this study assumes every rainfall pattern occurs equally; $p_i = 1/N$; thereby equation (4) is, in this study, a simple average of non-exceedance probability of rainfall. This assumption is verified by using a large number of rainfall patterns (1,584 patterns in this study).

## 2.4 Probability distribution of annual maximum flood peak discharge (AMF)

Given that occurrence of a storm event follows the Poison process with occurrence ratio $\mu_a$, the occurrence ratio of a flood event that causes flood peak discharge larger than $Q_p$ is:

$$\mu_Q = \mu_a\left(1 - G_{Q_p}(Q_p)\right) \tag{5}$$

Then, the CDF of AMF $F_{Q_p}(Q_p)$ is represented by the compound Poison distribution as follows:

$$F_{Q_p}(Q_p) = \exp\left[-\mu_Q \Delta t\right] = \exp\left[-\mu_a \Delta t\left(1 - G_{Q_p}(Q_p)\right)\right] \tag{6}$$

Since the sum of probability occurrence of each rainfall pattern $p_i$ is 1 (i.e. $\sum_{i=1}^{N} p_i = 1$), equation (4) can be rewritten as follows:

$$1 - G_{Q_p}(Q_p) = \sum_{i=1}^{N} p_i\left[1 - G_{R_a|D}\left(r_{a,i}(Q_p)\mid d_i\right)\right] \tag{7}$$

Thus, by substituting this equation into equation (6), $F_{Q_p}(Q_p)$ is obtained by the following equation;

$$F_{Q_p}(Q_p) = \exp\left[-\mu_a \Delta t \sum_{i=1}^{N} p_i\left(1 - G_{R_a|D}\left(r_{a,i}(Q_p)\mid d_i\right)\right)\right] \tag{8}$$

Finally, the CDF of AMF $F_{Q_p}(Q_p)$ is obtained with R-Q relationships for $i$-th rainfall pattern $r_{a,i}(Q_p)$ and the conditional probability distribution of rainfall on duration $G_{R_a|D}(r_a\mid d)$, which are estimated in 3.1 and 3.4, respectively. As shown in equation (8), the RFFM theoretically derives CDF of AMF and thus any Monte Carlo simulation is not required in the calculation procedure of the RFFM.

## 3 Application of the Rainfall-based Flood Frequency Model (RFFM)

The proposed method was applied to flood frequency estimations at the Hirakata discharge gauging station. Figure 4 shows an upstream area of the Hirakata station in the Yodo River basin, which is composed of three tributary river basins; Katsura, Uji and Kizu River basins. The Yodo River basin has seven dam reservoirs; the Hiyoshi, Amagase, Takayama, Nunome, Murou, Hinachi and Shorenji Dams. In addition, the Uji River has the Seta Weir on outflow of Lake Biwa: the Kasura and Kizu Rivers have the Kameoka and Ueno retarding basins respectively, which are often flooded during past heavy rainfall events, whereas the Kyoto City area (the rectangular red domain in Figure 4) is rarely flooded because protected by a complex river system with many dam reservoirs and upstream floodplains, but small areas were flooded by overflow by Typhoon No. 18, 2013 [The Ministry of Land, Infrastructure, Transportation and Tourist, 2013]. This typhoon is the largest





on record and thus the Kyoto City area has not experienced any larger flooding. In general, this type of highly regulated river basins hardly experience extreme floods that exceed the design level, frequencies of which are the main focus of this study. To estimate flood frequencies using equation (8), The following sub-section 3.1 obtains conditional CDF of rainfall on duration $G_{R_a|D}(r_a \mid d)$. Using a rainfall-runoff model and a flood-inundation model described in 3.2 and 3.3, sub-section 3.4 obtains R-Q relationships for various rainfall patterns $r_{a,i}(Q_p)$.

### 3.1 Estimation of conditional probability function of total basin rainfall on rainfall duration

To estimate the conditional CDF of total basin rainfall given duration $G_{R_a|D}(r_a \mid d)$, the joint CDF of total basin rainfall and rainfall duration $G_{R_a D}(r_a, d)$ is obtained by a copula approach [*Nelsen,* 2006]. A copula approach estimates the dependence structure of two variables by using copula functions. The joint CDF $G_{R_a D}(r_a, d)$ is obtained by a copula function $C$ and marginal distributions $G_{R_a}(r_a)$ and $G_D(d)$ as follows:

$$G_{R_a D}(r_a, d) = C\big(G_{R_a}(r_a), G_D(d)\big) \tag{9}$$

Copulas separate multivariate CDF and its marginal distributions. This study fitted basin rainfall and duration data separately to the exponential, gamma and Generalized Pareto (GP) distributions with the maximum likelihood method and selected the GP distribution by the Akaike information criterion (AIC) [*Akaike,* 1973]. Among a number of proposed copula functions, this study used a one-parameter family of copulas and selected the normal copula [*Nelson,* 2006] using a maximum likelihood method and the AIC. By using the normal copula, the joint CDF of rainfall and duration is defined as follows [*Nelson,* 2006]:

$$G_{R_a D}(r_a, d) = C\big(G_{R_a}(r_a), G_D(d)\big) = \Phi_\Sigma\big(\Phi^{-1}\big(G_{R_a}(r_a)\big), \Phi^{-1}\big(G_D(d)\big)\big) \tag{10}$$

where $\Phi_\Sigma$ is a two-dimensional standardized normal distribution; $\Phi$ is a one-dimensional standardized normal distribution. $\Sigma$ is the correlation matrix of $\Phi_\Sigma$, defined as follows:

$$\Sigma = \begin{pmatrix} 1 & \theta \\ \theta & 1 \end{pmatrix} \tag{11}$$

where $\theta$ is a parameter of the normal copula. Figure 5 shows the observed rainfall and duration of 1,500 samples and the results simulated by equation (10). The estimated joint CDF $G_{R_a D}(r_a, d)$ represents the characteristics of the observed scattered plot well. The Spearman's correlation coefficient for the observed data and the generated samples from the joint





probability distribution are 0.377 and 0.329, respectively. The conditional CDF of total basin rainfall on duration $G_{R_a|D}(r_a \mid d)$ is then obtained by the joint probability density function (PDF) $g_{R_a D}(r_a, d)$ and the marginal PDF of rainfall duration $g_D(d)$ as follows;

$$G_{R_a|D}(r_a \mid d) = \int_{-\infty}^{r_a} g_{R_a|D}(\tilde{r} \mid d) d\tilde{r} = \int_{-\infty}^{r_a} \frac{g_{R_a,D}(\tilde{r},d)}{g_D(d)} d\tilde{r} \tag{12}$$

## 3.2 Rainfall-runoff model

To obtain R-Q relationships for various rainfall patterns, a rainfall-runoff model was constructed in the Yodo River basin, which has steep and humid slopes, being located in mountainous terrain. Rainfall-runoff components in such basins are well represented by a kinematic wave-based rainfall-runoff model [*Sayama and McDonnell,* 2009; *Hunukumbura et al.,* 2012; *Tanaka and Tachikawa,* 2015]; thus, this study applies a kinematic wave-based distributed rainfall-runoff model 1K-DHM ver. 2 [*Tachikawa and Tanaka,* 2013] to the Yodo River basin. 1K-DHM ver. 2 is a distributed rainfall-runoff model based on a kinematic wave flow approximation that considers surface-subsurface flow. The elevation and the flow direction of each cell are determined using HydroSHEDS [*Lehner et al.,* 2006] with digital elevation models (DEMs) in 30 second (around 1 km) resolution. 1K-DHM ver. 2 consists of river cells with slope-runoff components. The schematic drawing of 1K-DHM ver.2 is shown in Figure 6. Each river cell has slope-runoff components on both sides, where runoff from slope-runoff components is simulated by the following discharge-storage relationship that considers surface-subsurface flow components [*Tanaka and Tachikawa,* 2015]:

$$q = \begin{cases} d_c k_c \left( \dfrac{h}{d_c} \right)^{\beta} i & \left( 0 \leq h \leq d_c \right) \\ d_c k_c i + \left( h - d_c \right) k_a i & \left( d_c \leq h \leq d_a \right) \\ \dfrac{\sqrt{i}}{n_s} \left( h - d_a \right)^m + \left( h - d_c \right) k_a i + d_c k_c i & \left( d_a \leq h \right) \end{cases} \tag{13}$$

In equation (13), $r$ is rainfall intensity, $h$ is water stage, $q$ is runoff per unit slope width, $d_c$ is an equivalent water stage to the maximum water content in the capillary pore, $k_c$ is hydraulic conductivity when the capillary pore is saturated, $\beta$ is an exponent parameter that describes the relationship between hydraulic conductivity and saturation, $k_a$ is saturated hydraulic conductivity (and $k_a = \beta k_c$ according to the continuity of the $q$-$h$ relationship shown in equation (13)); $d_a$ is the water stage equivalent to the maximum water content in the effective porosity, $n_s$ is the Manning's roughness coefficient for overland flow in the slope runoff cells; $i$ is slope gradient, and $m = 3/5$. Equation (13) represents the $q-h$ relationship for the





surface and subsurface soil layer. Water stage $h$ and discharge per unit slope $q$ are calculated by combining equation (13) with the continuity equation shown in equation (14):

$$\frac{\partial h}{\partial t} + \frac{\partial q}{\partial x} = r \tag{14}$$

River flow is simulated by the following 1-D kinematic wave equations.

$$Q = \alpha A^m \tag{15}$$

$$\frac{\partial A}{\partial t} + \frac{\partial Q}{\partial x} = 2q_s \tag{16}$$

where $Q$ is the river discharge, $A$ is the cross-sectional area (and $\alpha = \sqrt{i} / (nB^{2/3})$), and $q_s$ is runoff from the slope-runoff component on each side of a unit cell.

The model parameters in equation (13) were calibrated to flood hydrographs during Typhoon No. 18 (2013) observed at dam stations that do not have any upstream dams; the Hiyoshi Dam in the Katsura River basin, the Amagase Dam in the Uji River basin, and the Murou Dam in the Kizu River basin. Typhoon No. 18 caused huge peak discharge at the Hirakata station and, damaging several downstream regions [*Kinki Regional Bureau, MLIT*, 2013]. Calibrated hydrographs for each of the three dam stations are shown in Figure 7. The simulated hydrographs agreed well with observed discharge for all stations. Inflow discharge observed in the Amagase Dam station (Figure 7b) is affected by release from the Seta Weir 40 hours after flooding. As this study focuses on flood peak discharge, dam release after flooding was not considered in the model. Calibrated parameters within the model are shown in Table 1.

According to dam operation rules in Japan, outflow discharge from upstream dams in the Yodo River basin was modelled as follows;

1. Dam inflow $I$ is released while $I$ is smaller than a pre-determined flood discharge $Q_f$;

2. Flood control starts when inflow discharge $I$ exceeds $Q_f$;

3. If dam storage volume $S_r$ reaches the effective total storage $S_{SL}$, all inflow $I$ is released to avoid overtopping;

4. After inflow peak time $T_{max}$, outflow discharge at peak time $I(T_{max})$ is released until dam storage volume $S_r$ is





reduced to the control storage volume $S_{NL}$ ;

According to the above rules, outflow discharge at time $t$ , $Q(t)$ is summarized as follows;

$$Q(t) = \begin{cases} I(t) \left( I(t) < Q_f \right) \\ a\{I(t) - Q_f\} + Q_f \left( I(t) \geq Q_f \text{ and } S_r < S_{SL} \right) \\ I(t) \left( I(t) \geq Q_f \text{ and } S_r \geq S_{SL} \right) \\ a\{I(T_{max}) - Q_f\} + Q_f \left( t > T_{max} \text{ and } S_r > S_{NL} \right) \end{cases} \qquad (17)$$

where $a$ is a constant value to determine outflow ( $a \leq 1.0$ ). Operations of upstream seven dams in the Yodo River basin were modeled by equation (17), setting the parameters $a$ , $Q_f$ , $S_{SL}$ and $S_{NL}$ according to the actual flood operation rule of each upstream dam.

The Seta Weir, located at the outlet of Lake Biwa, is closed during severe flooding; thus, outflow from the weir was set to zero during floods in the model. At the Kameoka and Ueno retarding basins, flood storage was modeled using a liner reservoir model;

$$\frac{dS_f}{dt} = I - Q \qquad (18)$$

$$S_f = k_f Q \qquad (19)$$

where $S_f$ is storage in a floodplain area, $I$ is inflow discharge to the floodplain area, $Q$ is outflow discharge from the floodplain area, and $k_f$ is a fitting parameter with units of time. $k_f$ of the Ueno retarding basin was calibrated to observed storage in Typhoon No. 18. $k_f$ of the Kameoka retarding basin (see Figure 4) was calibrated to reproduce flood peak discharge at the Hirakata station. The simulated flood peak discharge from Typhoon No. 18 (shown in Figure 8) was 9,900 $m^3/s$, which agreed with the estimated flood peak discharge of 9,500 $m^3/s$ by *Kinki Regional Bureau, MLIT,*[2013]. The calibrated 1K-DHM model was validated using the observed discharge data during the Isewan Typhoon, which caused severe flood damage in large areas including the Kyoto city area in 1959. The simulated hydrograph of this typhoon event is shown in Figure 9. In 1959, the seven current dams were not constructed; additionally, the Seta Weir was functional, and the two floodplains largely flooded during this time. Therefore, the six dam reservoirs were excluded from simulations, which resulted in peak discharge of 8,000 $m^3/s$ in close agreement with the peak discharge of 8,099 $m^3/s$.



### 3.3 Flood inundation model

To obtain R-Q relationships with consideration of inundation in an upstream area, a large number of flood-inundation simulations are required. In river basins in Japan, inundation occurs within limited floodplains due to their steep mountainous topography; therefore, we applied an inundation model coupled with the rainfall-runoff model 1K-DHM ver. 2

(hereinafter referred to as "IMCR") to the Kyoto city area where relatively large extent of inundation is expected during extremely floods. Although a number of recent studies [Kobayashi and Takara, 2013; Sayama et al., 2012] simulate flood-inundation within the entire river basin, the IMCR simulates 2-dimensional flood propagation in selected floodplain areas, provided upper and lateral boundary river discharges by 1K-DHM ver. 2 through an automatic connection system described in Figure 10, which enables a large number of flood inundation simulations with small computational expense.

First, 1K-DHM ver. 2 simulates runoff from all cells (the cells in the upper figure) and river routing according to flow direction (the arrows); then simulated river discharges by 1K-DHM ver. 2 flowing into the IMCR domain (the red arrows) are provided with IMCR river cells (the blue cells). Setting boundary river discharge provided by 1K-DHM, the IMCR simulates river routing along the IMCR river cells and flood-inundation on IMCR floodplain cells (the green cells) when river water stage exceeds river bank heights. There are two ways to provide boundary river discharge from 1K-DHM ver. 2

to the IMCR; 1) if the most upstream IMCR river cell is located on the boundary of the IMCR domain (the yellow cells), it takes river discharge from the upstream 1K-DHM cell (e.g. $Q_{U1}$ and $Q_{U2}$ from the light blue cells); 2) if IMCR river cells are not located at the boundary of the corresponding 1K-DHM cell and discharge from another 1K-DHM cell (the orange cell) flows into the 1K-DHM cell, river discharge from the orange 1K-DHM cell   is provided to the all IMCR river cells as lateral inflow, equally divided by the number of the IMCR river cells (one over forth in Figure 10). This coupling system

enables automatic connection between the two models. To focus on changes of river discharge by river overflow, the IMCR only considers inundation by river water, which assumes that floodwater returns to river cells only through overflow from inundation cells to river cells.

IMCR uses local inertial equations to simulate flood inundation at a smaller computational cost [*Bates et al.,* 2010]. The IMCR consists of river cells and inundation cells. River routing cells are calculated by 1-D local inertial equations:

$$\frac{\partial Q}{\partial t} + gA\frac{\partial(h+z)}{\partial x} + \frac{gn^2|Q|Q}{R^{4/3}A} = 0 \tag{20}$$

$$\frac{\partial A}{\partial t} + \frac{\partial Q}{\partial x} = q_{L} - q_{o} \tag{21}$$





where $Q$ is the river discharge, $h$ is the water depth, $z$ is the elevation, $A$ is the flow cross section area, $R$ is the hydraulic radius, $n$ is Manning's roughness coefficient, $q_L$ is lateral inflow provided by 1K-DHM, and $q_o$ is overflow discharge from river cells per unit cell width. $A$ and $R$ depend on water depth $h$, according to $A = f_A(h)$, $R = f_R(h)$. $q_o$ is calculated by the momentum equation of local inertial equations between neighboring river and inundation cells. Using the

following 2-D local inertial equations, river water inundation into adjacent cells was simulated:

$$\frac{\partial q_x}{\partial t} + gh\frac{\partial(h+z)}{\partial x} + \frac{gn_f^2|q_x|q_x}{h^{7/3}} = 0 \tag{22}$$

$$\frac{\partial q_y}{\partial t} + gh\frac{\partial(h+z)}{\partial y} + \frac{gn_f^2|q_y|q_y}{h^{7/3}} = 0 \tag{23}$$

$$\frac{\partial h}{\partial t} + \frac{\partial q_x}{\partial x} + \frac{\partial q_y}{\partial x} = q_o \tag{24}$$

where $n_f$ is Manning's roughness coefficient of floodplain cells, and $q$ is river discharge per unit cell width.

Relations between water depth and cross-sectional area $f_A(h)$, and between water depth and the hydraulic radius $f_R(h)$ were obtained from cross-section river measurements from the main stem and three tributary rivers. The Manning's roughness coefficients for the main channel and floodplains on the left and right sides ($n_c$, $n_l$ and $n_r$) are provided by the Kinki Regional Development Bureau, MLIT. Using them, the roughness coefficients of river channel cells in the IMCR $n$ were calculated using Lotter's Method, [*Lotter,* 1933] as follows:

$$n = \frac{R^{2/3}A}{\frac{1}{n_c}i^{1/2}R_c^{2/3}A_c + \frac{1}{n_l}i^{1/2}R_l^{2/3}A_l + \frac{1}{n_r}i^{1/2}R_r^{2/3}A_r} \tag{25}$$

where hydraulic variables with suffixes of c, l, and r indicate those describing the main channel and floodplains to the left and right of the river. The Manning's roughness coefficient of floodplain cells $n_f$ was set to 0.05, which is greater than that of the main channel, which ranges from 0.02 to 0.045.



### 3.4 Relationships between total basin rainfall and flood peak discharge

Relationships between total basin rainfall and flood peak discharge at the Hirakata gauging station (R-Q relationships) for various spatial-temporal distribution patterns of rainfall events were calculated using 1K-DHM ver. 2 and the IMCR calibrated above. To see the impact of upstream dam operation and flooding in the Kyoto City area, R-Q relationships were

5 obtained for three cases: without dam operation and without an inundation model (WOD/WOI), with dam operation and without an inundation model (WD/WOI), and with dam operation and with an inundation model (WD/WI). For each case, R-Q relationships were obtained through the following steps:

1) To prepare rainfall patterns $\xi_i(x, y, t)$ $(i = 1,2,..., N)$ from observed $N$ rainfall events

2) To create a storm event with total basin rainfall $r$ and rainfall pattern $\xi_i(x, y, t)$, by increasing total basin rainfall of a

rainfall event fixing its spatial-temporal distribution $\xi_i(x, y, t)$

3) To simulate flood peak discharge using 1K-DHM ver. 2 from the created storm event

4) To increase total basin rainfall $r$ and repeat the step 3)

For the case WD/WI, simulated river discharge using 1K-DHM ver. 2 in the step 3) was connected to the IMCR and it simulated flood peak discharge at the Hirakata station considering inundation within the upstream Kyoto City area (the

15 rectangular red domain in Figure 4). The R-Q relationships WOD/WOI, WD/WOI and WD/WI are shown in Figure 11. Rainfall patterns of the past 1,548 rainfall events are prepared by spatial interpolation, using the Thiessen polygon method, for data from 156 gauging stations (Figure 4) from 1980 to 2014. All Figures indicate that R-Q relationships vary significantly across the range of rainfall patterns. The RFFM considers variability in the spatial-temporal distribution of rainfall empirically by using as many rainfall patterns as possible. Figure 11 clearly shows different characteristics. The

20 WOD-WOI case (Figure 11a) shows the largest peak discharge for a particular rainfall among all three cases; the WD-WOI case (Figure 11b) shows slightly smaller discharge than the WOD-WOI for the same rainfall because of the effect of upstream dam operation but the shapes of R-Q relationships in Figures 11a and 11b are similar. In Figure 11c, gradient of the R-Q relationships for all of the rainfall patterns is much smaller than other cases at larger flood peak discharges than 13,000 m³/s. This is because inundation occurs in the Kyoto City area when flood peak discharge of the Hirakata station is over

25 13,000 m³/s in the IMCR with measured river cross-sectional data. Observed overflow inundation in the largest flood event in 2013 supports this. It indicates that, in the Yodo River basin, the inundation in the Kyoto City area has a larger impact on flood peak discharge at the Hirakata station than flood control by upstream dams. This is because, as seen in Figures 7 and 8, flood peaks at the upstream dams are much smaller than that at the Hirakata station. The R-Q relationships for all rainfall





patterns $r_{a,i}(Q_p)$ and the conditional probability distributions of basin rainfall on duration $G_{R_a|D}(r_a \mid d)$ obtained above are substituted into equation (8) to calculate CDF of AMF $F_{Q_p}(Q_p)$ at the Hirakata station.

## 4 Results and discussions

### 4.1 Frequency analysis of annual maximum flood peak discharge data

To validate the RFFM estimations, this section applied conventional FFA. To eliminate the uncertainty of rainfall-runoff modelling, the time series of past discharges simulated using 1K-DHM ver. 2 (and the IMCR for WD/WI) was used for FFA, instead of observed discharge data. Figure 12 shows annual maximum flood peak discharge WOD/WOI, WD/WOI, and WD/WI. The three estimations are quite similar to each other. Note that the effect of dam operation (difference between red and green bars in Figure 12) is larger than that of inundation of the Kyoto City area (difference between green and blue bars in Figure 12) for large flood events (1983 and 2013) and large inundation was not observed. Small differences between WD-WI and WD-WOI are caused by differences between kinematic wave-based river routing in 1K-DHM ver. 2 and river routing based on the local inertial equations in the IMCR. Results for all the three cases were fitted to a generalized extreme value (GEV) distribution [*Stedinger et al.*, 1993] using the maximum likelihood method.

Figure 13 shows flood peak discharge for all previous rainfall events. The POT approach has been also applied to each of the three data sets. The selected threshold value of the POT approach needs to be large enough to extract extreme values and at the same time be small enough to have sufficient samples. In this study, the threshold value is determined as the top 1% of samples; 1,016.66 m$^3$/s, 1,196.63 m$^3$/s and 1,212.16 m$^3$/s for WD/WI, WD/WOI and WOD/WOI, respectively. The number of samples over the threshold is 66 in all the three cases. Each sample set above the threshold value was applied to the GP distribution [*Stedinger et al.*, 1993] using the maximum likelihood method.

### 4.2 Comparison of flood frequency estimations by the RFFM and FFA

The estimated return periods of AMF at the Hirakata station by the RFFM and FFA (AMS and POT approaches) without consideration of upstream dams and inundation in the Kyoto City area (WOD/WOI) are shown in Figure 14. The dots are estimated quantiles using the Cunnane plotting positions. The results by RFFM and FFA show slightly different return periods over 100 years but both are in a good agreement with those by the Cunnane plotting position formula. The RFFM does not use discharge data, which converts the CDF of total basin rainfall into that of AMF by the proposed probabilistic modelling. Figure 14 indicates that the probabilistic modelling of the RFFM successfully represents the probabilistic structure of basin rainfall with respect to flood frequency estimations. Figure 15 shows flood frequencies with upstream dam control and without inundation in the Kyoto City area (WD/WOI). In Figure 15, return periods estimated by both the RFFM and FFA are slightly larger than those in Figure 14 (e.g. a return period of flood peak discharge 10,000 m$^3$/s is 62 years in



Figure 14 and 104 years in Figure 15). This indicates that the effect of dam operation on downstream flood frequencies is successfully represented by the both approaches.

Figure 16 shows the relationship between flood peak discharge and its return period when both upstream dam operation and inundation of the Kyoto City area are taken into account (WD/WI). In Figure 16, all approaches show similar return period

estimations up to 13,000 m$^3$/s; however, beyond this, return periods of flood peak discharge estimated by the RFFM drastically increase, whereas the results by FFA does not show such drastic change. This drastic change of extreme flood frequencies is represented the drastic decrease of the corresponding flood peak discharge in R-Q relationships (see Figure 11 (C)). This impact was not observed in past discharge data; thus, FFA based on past discharge data does not represent drastic return period changes due to upstream inundation for extremely large discharge. On the other hand, the effect of upstream

dam operation reflected on the result that flood frequencies are smaller in Figure 15 than Figure 14. This is because the effect of dam operation appears in past discharge data.

This experiment clarified that FFA represents probabilistic structure of observed data but difficult to extrapolate drastic changes of flood frequencies which the river basin has not experiences. On the other hand, the RFFM represents both flood frequencies with various return periods. This study demonstrated the limitation of data-based FFA and the advantage to

apply the proposed RFFM to flood frequency estimations of highly regulated river basins where floods that exceed the design level rarely occur.

**5 Conclusions**

    The estimation of T-year flood is essential to design flood defenses. In addition, considering adaptation measures against devastating floods that exceed the design level, frequency estimations of extreme floods are essential. To achieve this, this

study introduced the rainfall-based flood frequency model (RFFM), which converts a probability distribution of total basin rainfall into that of annual maximum flood peak discharge (AMF) by probabilistic rainfall modelling directly using observed spatial-temporal rainfall profiles. The RFFM was applied to the Hirakata discharge gauging station in the Yodo River basin (7,290 km$^2$), Japan. The estimated flood frequencies by the RFFM were compared to those of flood frequency analysis (FFA) of the time series of discharge data. From the results, we obtained the following findings:

1.   The estimated probability distribution of AMF by probabilistic rainfall modelling with empirical representation of spatial-temporal rainfall pattern well reproduced that from FFA of discharge data, for the river discharge that the station has experienced;

    2.   Statistical changes in AMF by the effect of upstream dam operation, which was observed in past discharge data, were successfully represented by both the RFFM and FFA;





3.  The return periods of food peak discharge over 13,000 m³/s estimated using the RFFM are much larger than those from FFA. The RFFM is able to represent the effect of inundation in upstream areas by using physical flood-inundation models, while data-based FFA is difficult to consider such impact that was not observed in the past.

FFA of discharge sample data more directly estimates flood frequencies than rainfall-based approaches; however, rainfall-based approaches have the advantage that drastic changes in statistical characteristics can be easily taken into account by physical rainfall-runoff modeling and extension of total basin rainfall with appropriate probabilistic rainfall modeling. This is also a type of extrapolation but based on physical rainfall-runoff modeling; thus, the RFFM has the potential for rational extrapolation of large discharges that a river basin of interest has not yet experienced, which is essential to design countermeasures against a changing climate and/or various changes in the river basins. We hope that the framework proposed in this study could help to estimate return periods of large river discharges that exceed design levels.

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



**Table 1: 1K-DHM parameters for each tributary basin (those in the Katsura River basin are applied to a downstream area of the confluence of the three tributary rivers)**

| Parameter | Katsura River Basin | Uji River Basin | Kizu River Basin |
|---|---|---|---|
| $n_s \left(\mathrm{m}^{-1/3}\mathrm{s}\right)$ | 0.40 | 0.10 | 0.54 |
| $k_a \left(\mathrm{m/s}\right)$ | $5.0 \times 10^{-4}$ | $3.0 \times 10^{-5}$ | $3.0 \times 10^{-4}$ |
| $d_a \left(\mathrm{m}\right)$ | 0.38 | 1.4 | 0.70 |
| $d_c \left(\mathrm{m}\right)$ | 0.16 | 0.40 | 0.50 |
| $\beta \left(-\right)$ | 6.0 | 6.0 | 7.0 |





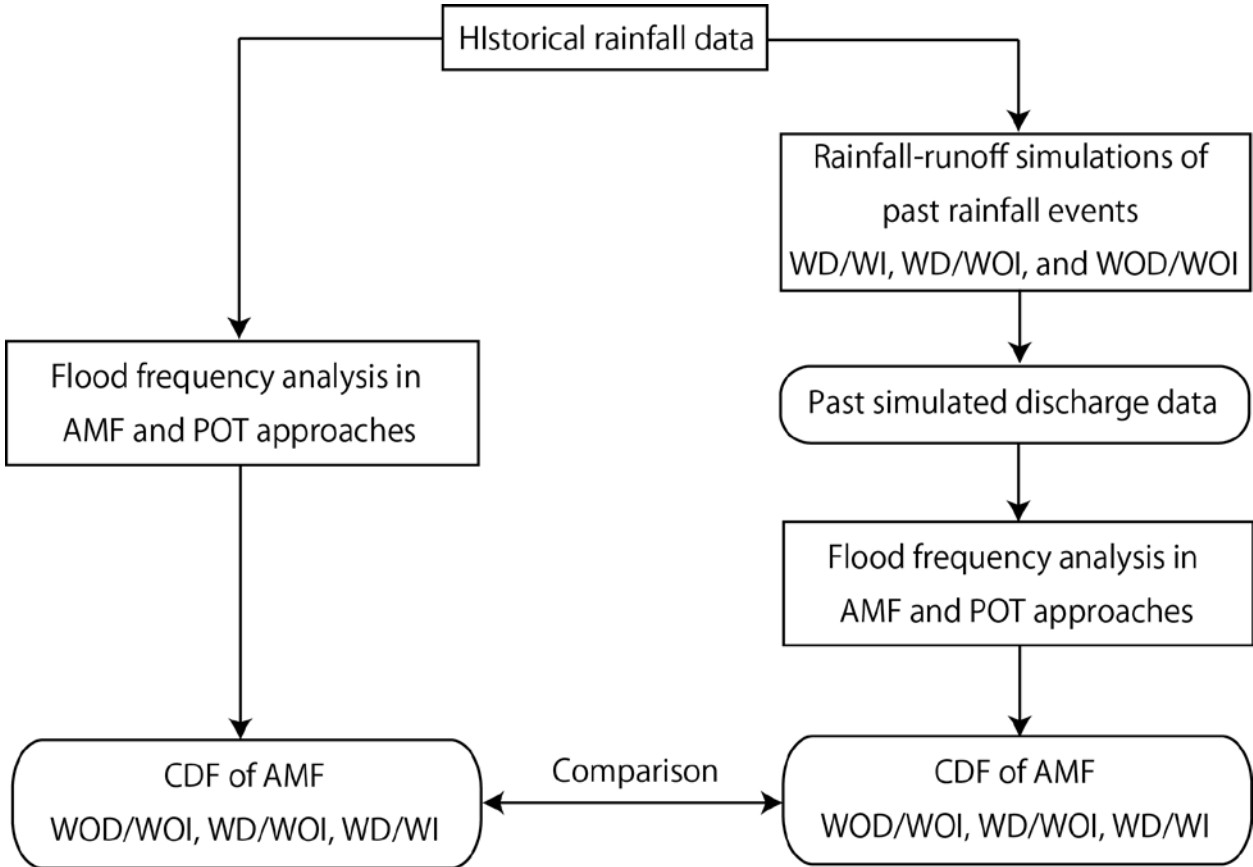

Figure 1: Schematic diagram of the study method. Flood frequencies are estimated by the rainfall-based flood frequency model (RFFM) and by FFA in three conditions; with dam operation and an inundation model (WD/WI), with dam operation and without an inundation model (WD/WOI) and without dam operation and an inundation model (WOD/WOI). Rounded and ordinary rectangles show products and processes.

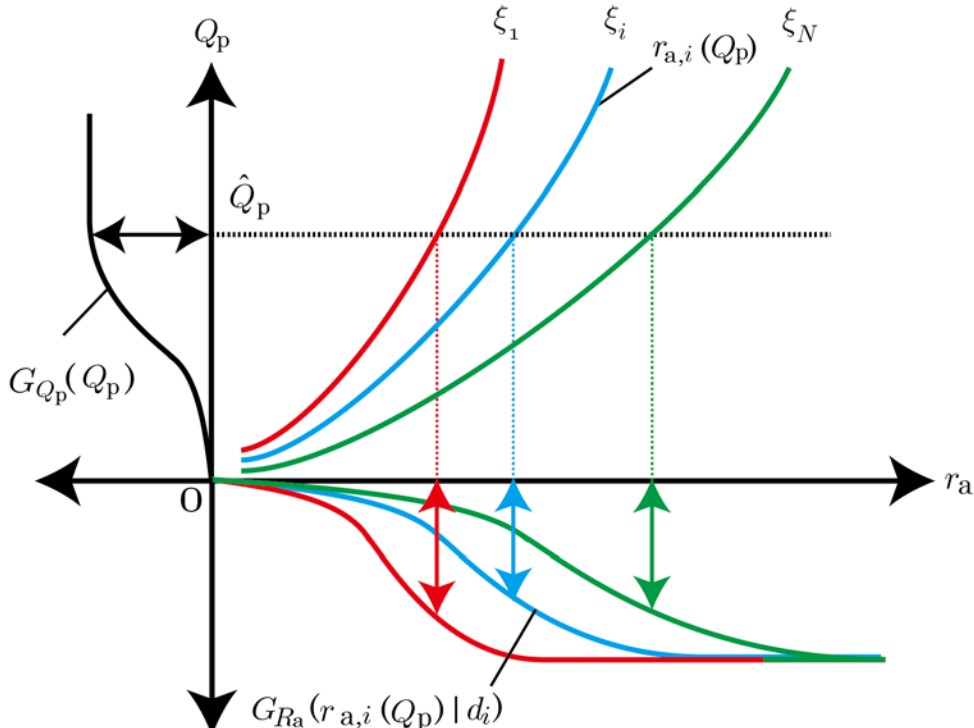

**Figure 2:Schematic diagram of calculation process of the cumulative distribution function (CDF) of the economic damage for one rainfall event by using the equation (4).**



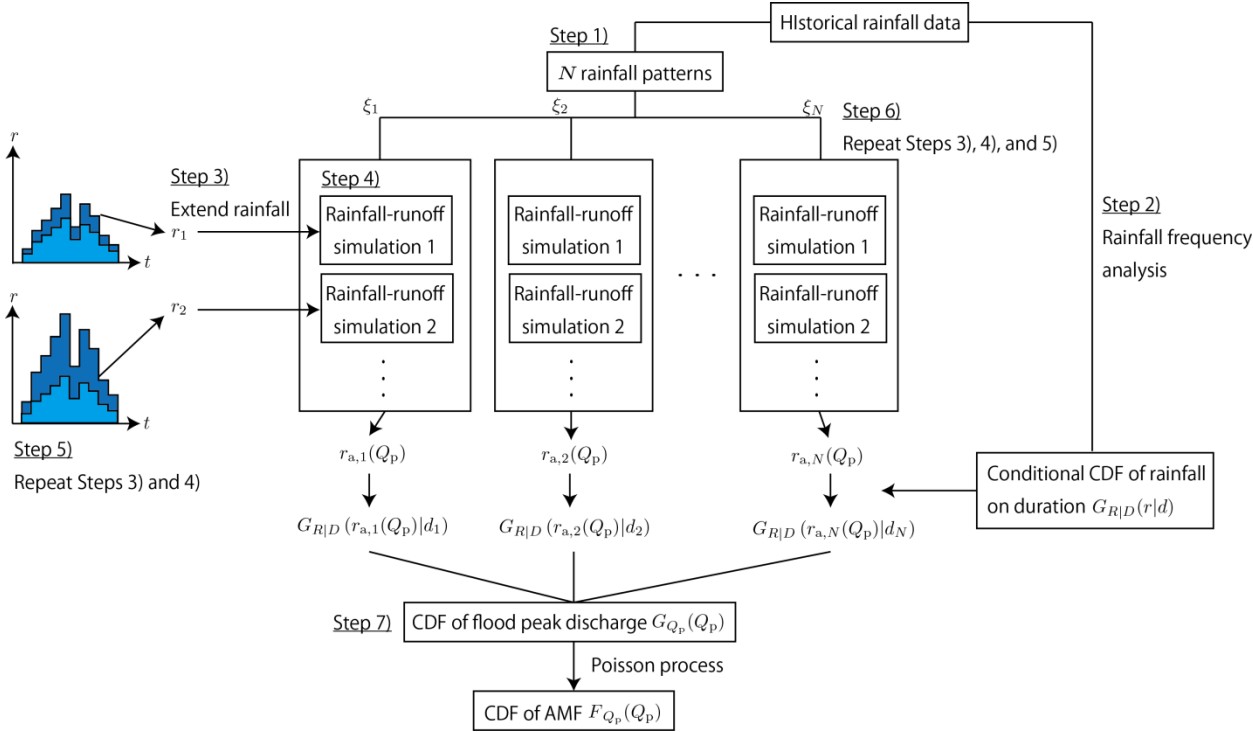

**Figure 3: Flowchart of a rainfall-based flood frequency model (RFFM). Steps 1) to 7) correspond to those in 2.3.**





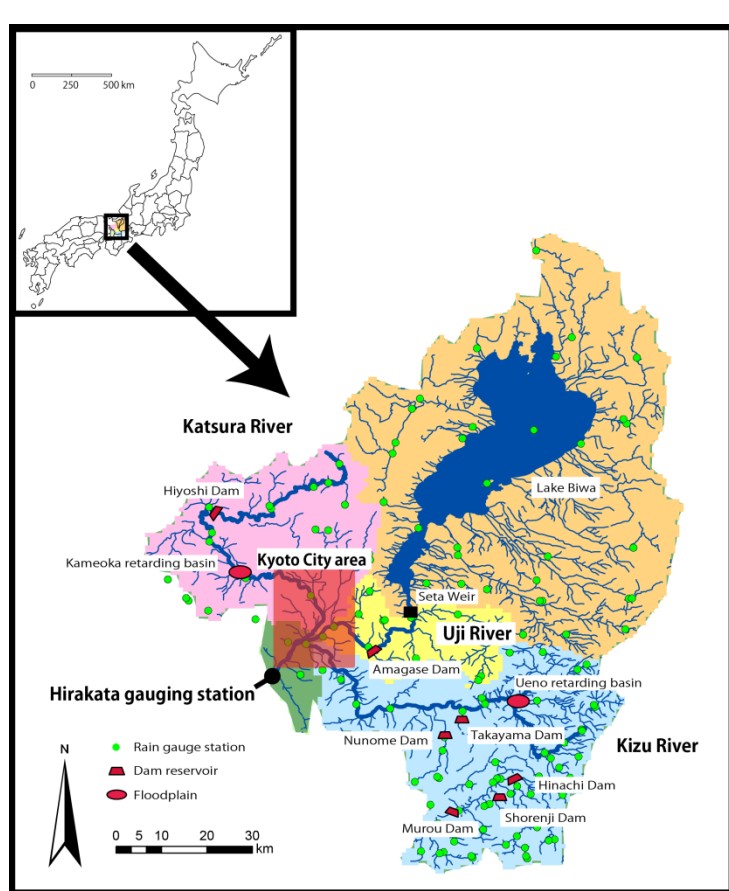

**Figure 4: An upstream area of the Hirakata discharge gauging station in the Yodo River basin. The black point is the Hirakata station; the thin blue lines are small river channels; the thick blue lines are major tributary rivers and the main rivers; the red trapezoids are dam reservoirs; the red areas are floodplain areas (the Kameoka and Ueno retarding basins and the Kyoto City area); the black square is the Seta Weir; and the green points are rain gauge stations.**





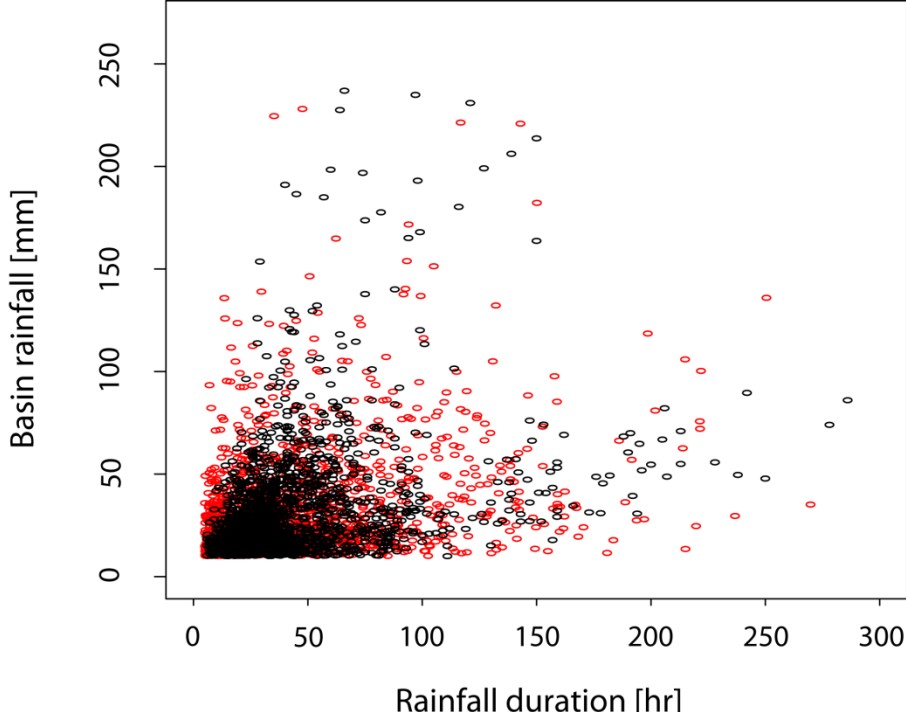

Figure 5: 1,500 samples of observed (black) and simulated (red) total basin rainfall and rainfall duration.



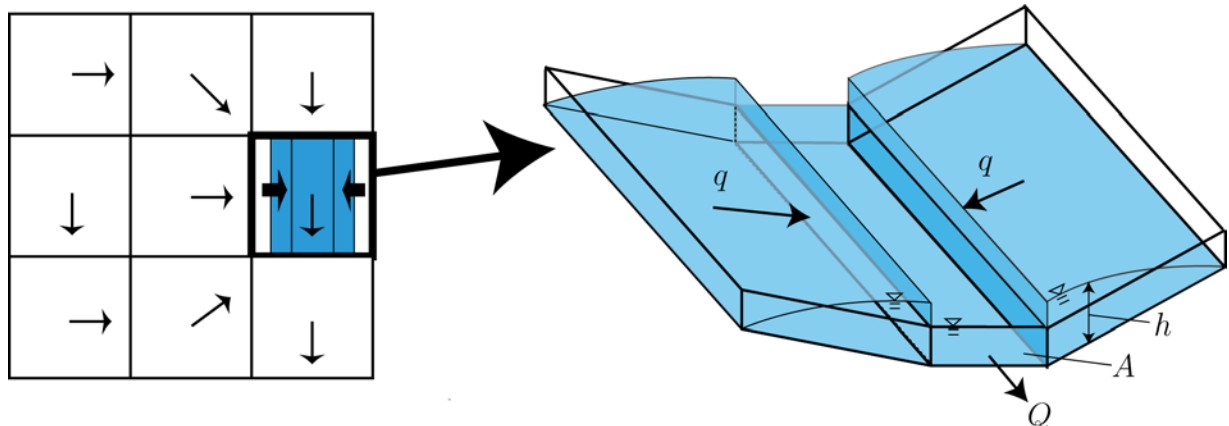

**Figure 6: Schematic diagram of 1K-DHM ver. 2**




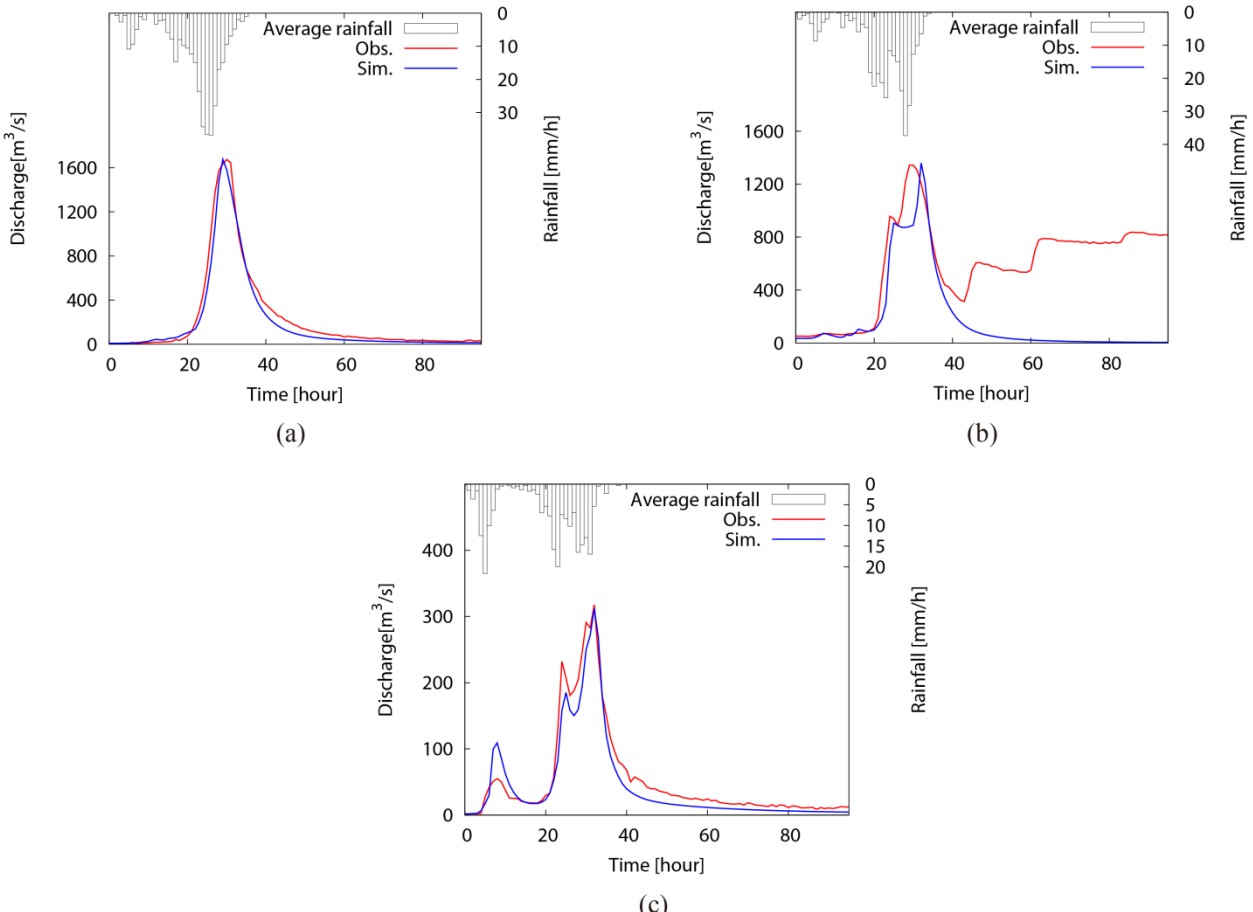

**Figure 7: Calibration results for the (a) Hiyoshi, (b) Amagase and (c) Murou Dam stations during Typhoon No. 18 (in 2013). The blue line is the simulated hydrograph, the red line is the observed hydrograph and the bar graph shows hourly rainfall.**





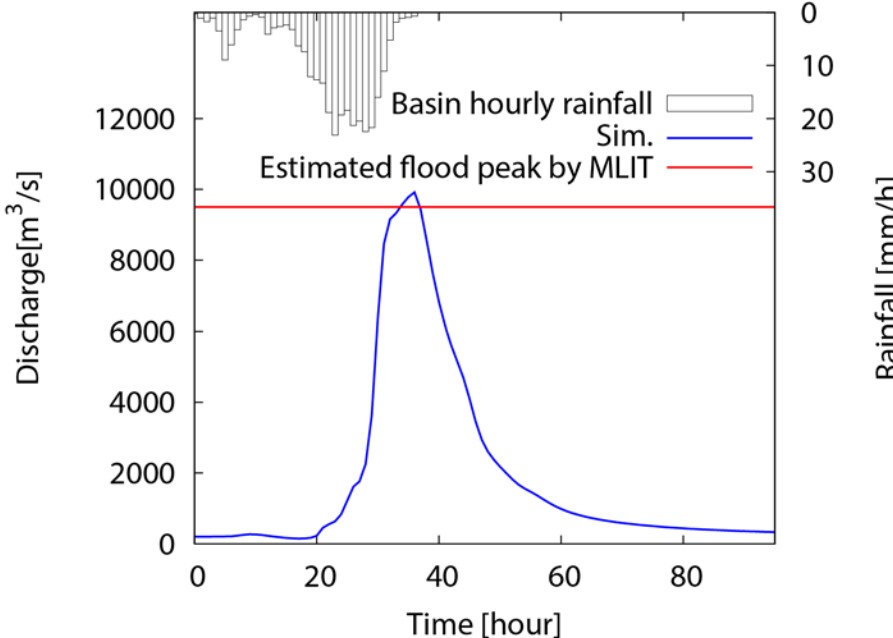

**Figure 8: The calibrated hydrograph for the Hirakata station during Typhoon No. 18, 2013. The blue line shows the simulated hydrograph, the red line shows the estimated flood peak discharge by Ministry of Land, Infrastructure, Transportation and Tourism (MLIT), and the bar graph shows hourly rainfall intensity.**




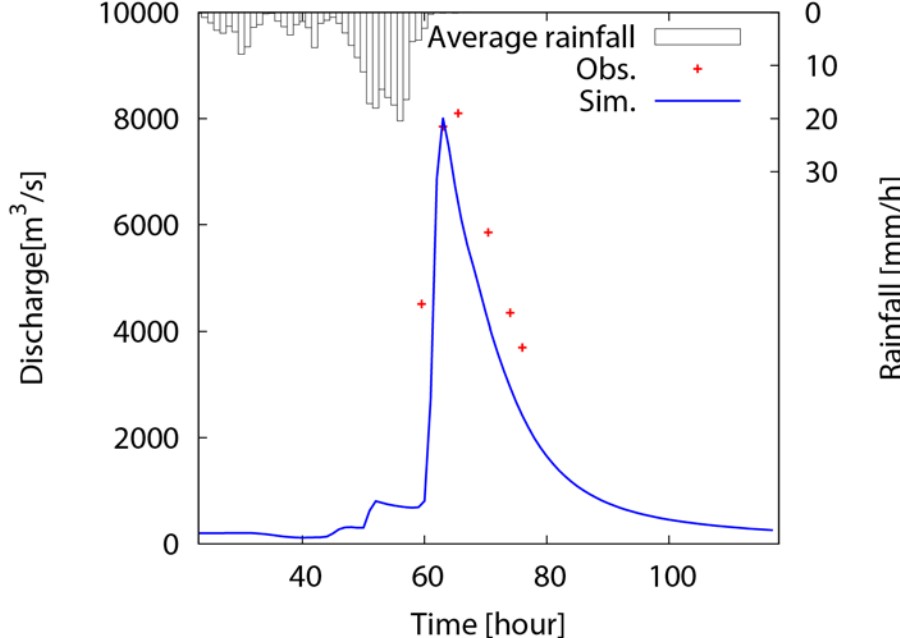

**Figure 9: Validation result for the Hirakata station during Isewan Typhoon in 1959. The blue line shows the simulated hydrograph, the dots show the observed discharge data, and the bar shows hourly rainfall intensity.**



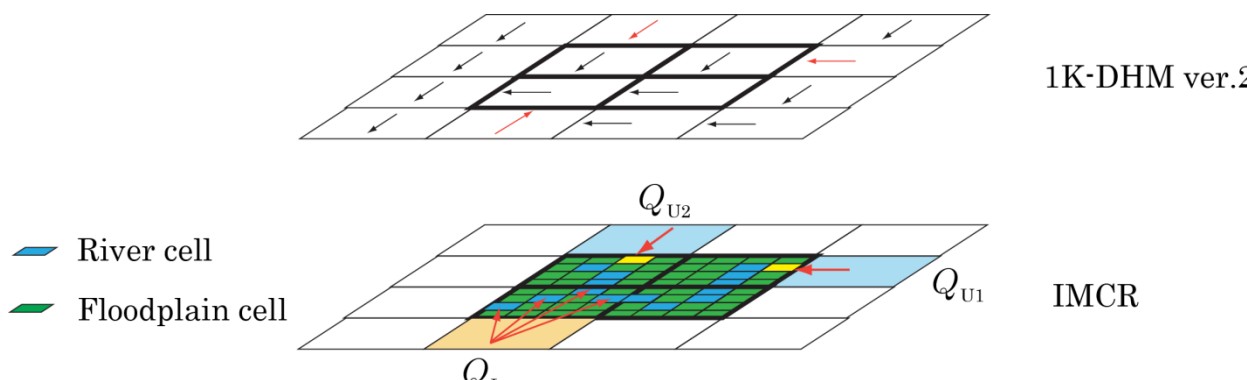

**Figure 10: Relationship between 1K-DHM and the IMCR. The thick cells show the 1K-DHM cells that cover to the IMCR domain; the coarser and finer cells show 1K-DHM and IMCR cells; the blue and green cells in the IMCR domain show river and floodplain cells of the IMCR.**




Figure 11: R-Q relationships at the Hirakata station for 1,548 rainfall patterns (a) WOD/WOI, (b) WD/WOI and (c) WD/WI





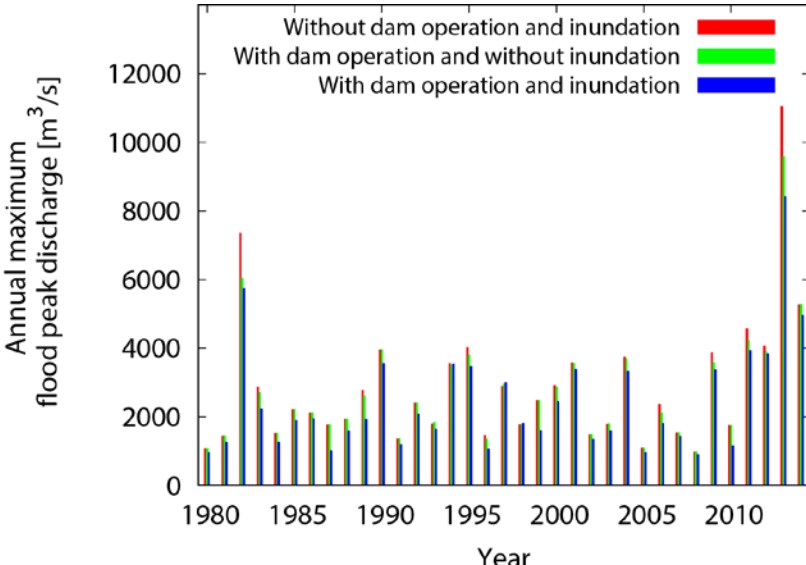

**Figure 12: Annual maximum flood peak discharge at the Hirakata station using WOD/WOI (red), WD/WOI (green) and WD/WI (blue)**





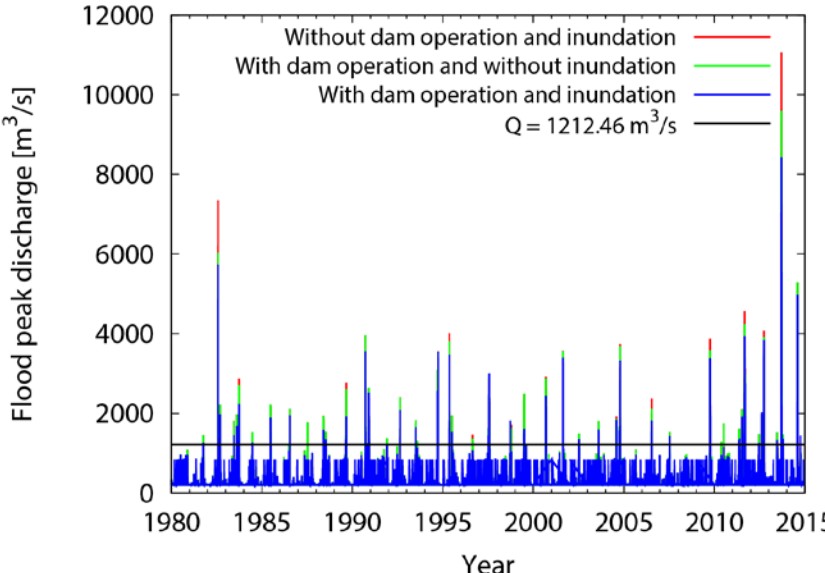

**Figure 13: Time series of flood peak discharge using WOD/WOI (red), WD/WOI (green) and WD/WI (blue) at the Hirakata station by all the past rainfall events. A black line shows a threshold value of the peak-over-threshold (POT) approach for the time series WD/WI.**





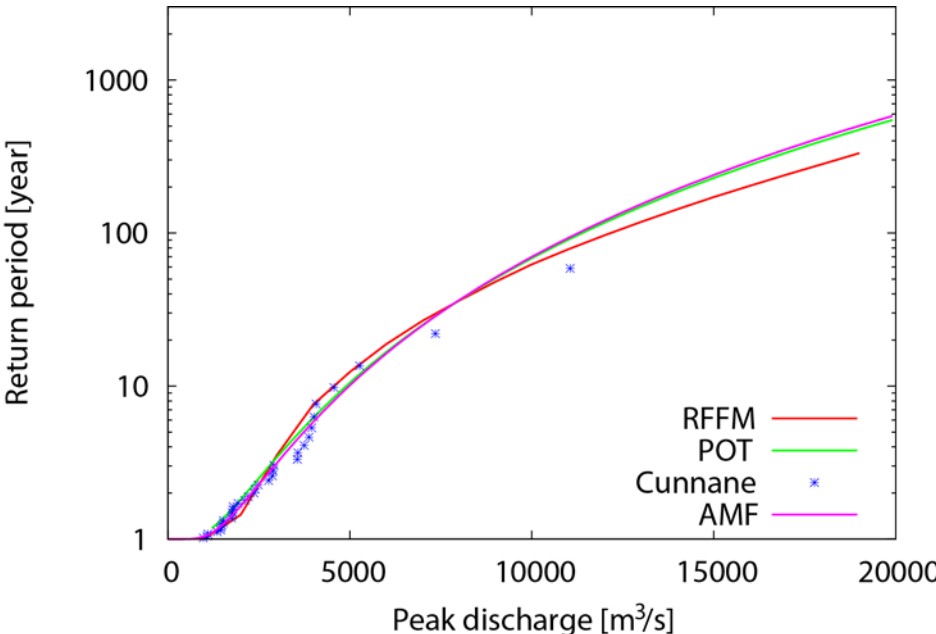

**Figure 14: Relationship between flood peak discharge and its return period at the Hirakata station estimated by the RFFM and FFA with annual maximum series and peak-over-threshold (POT) approaches (without consideration of flood control in upstream dams and inundation modeling of the Kyoto City area). Dots are the Cunnane plotting positions.**





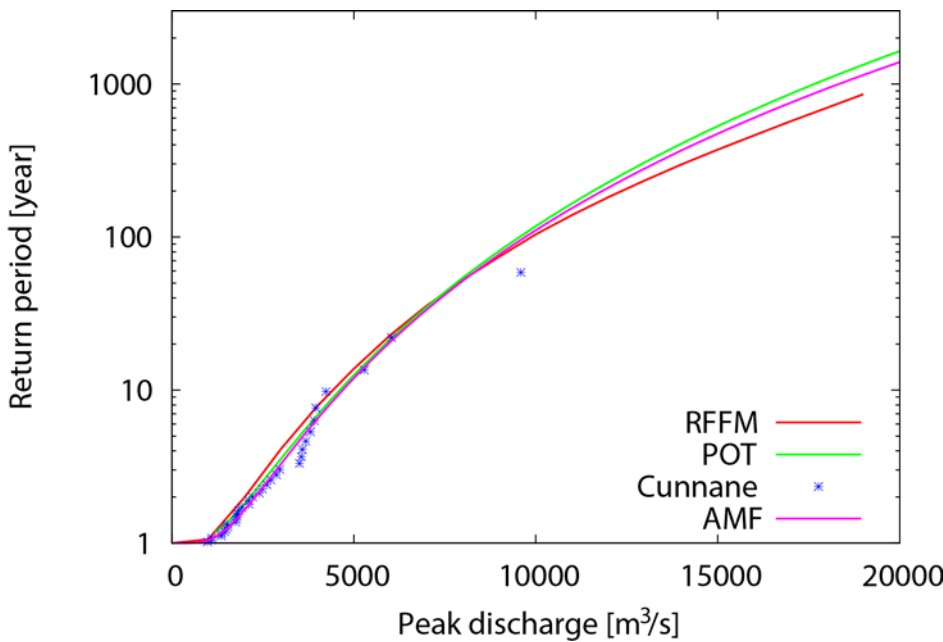

**Figure 15: Relationship between flood peak discharge and its return period estimated by the RFFM and FFA with annual maximum series and peak-over-threshold (POT) approaches (with consideration of upstream dams and without consideration of inundation modeling of the Kyoto city area). Dots are the Cunnane plotting positions.**





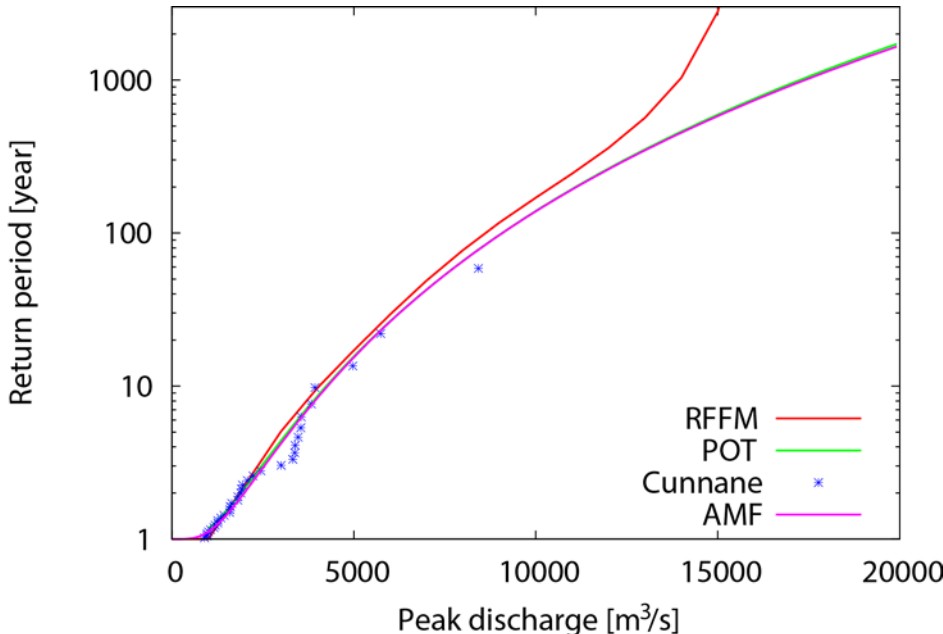

**Figure 16: Relationship between flood peak discharge and its return period estimated by the RFFM and FFA with annual maximum series and peak-over-threshold (POT) approaches (with consideration of flood control in upstream dams and inundation modeling of the Kyoto city area by the IMCR). Dots are the Cunnane plotting positions.**