# Peer review of "Frequency analysis of extreme floods in a highly developed river basin"

_Hydrology and Earth System Sciences, 2016_

## Referee Comment (RC1) · Anonymous Referee #1 · 22 Jun 2016

The manuscript describes an application of flood frequency analysis in a large river basin. Two methodologies are compared: the RFFM model based on rainfall data, and the standard FFA approach based on discharge values (or better on simulated data).

Although the topic is interesting, there are several drawbacks in the manuscript that do not allow me to suggest its publication.

The main issue is that it is not clear the innovative contribution, indeed the RFFM model refers to a previous publication of the authors (Tanaka et al. 2015a) so, apparently, the present manuscript provides only a well developed case study.

From the methodological point of view, I am skeptical on the adoption of RFFM model. Which is the added value of this procedure compared to the FFA applied on runoff simulations? In my opinion a continuous rainfall-runoff modelling can provide more useful

and complete information (indeed it would give also the entire design hydrograph).

Minor comments

page 1 line 22: Not only with AMF in general with extreme value sampling procedures.

page 2 line 6. The added value in considering rainfall data is mainly because these data are largely available compared to runoff data.

page line 17. Literature is very large about this topic, there are many other options not mentioned in the literature review.

page 6 line 9. Poisson instead of Poison

page 7 line 15. Nelsen not Nelson (this typo is present also in other part of the text)

page 7 line 15. It is not clear why the Normal copula was adopted.

In the Introduction authors say that they will compare RFFM approach to observed data, however at the end they use simulated data. I would be consistent and already in the introduction I would say exactly what will be compared at the end.

---

## Author Comment (AC1) · 5 Jul 2016

[Comment]

*The main issue is that it is not clear the innovative contribution, indeed the RFFM model refers to a previous publication of the authors (Tanaka et al. 2015a) so, apparently, the present manuscript provides only a well developed case study.*

[Response]

Thank you very much for your comment. As you pointed out, the framework of this study is based on the method the authors have proposed in the journal of Japan Society of Civil Engineering (JSCE). We think that this study is quite important as a case study, which demonstrated that for rationally extrapolating extreme flood frequencies, we have to be careful for hydrologic phenomena which have not occurred ever. As an example, this study focused on two particular phenomena: upstream inundation and dam operation. With regard to upstream inundation, because the target basin has not experienced it, conventional flood frequency analysis (FFA) from discharge samples of past storm events did not explain drastic decrease in frequencies of extreme floods, whereas the effect of dam operation was represented in conventional FFA because it has been observed in past flood events. This perspective is helpful for discussing the balance of flood risk in upstream and downstream areas. The revised manuscript will more clearly state the importance of the above.

In addition, although the basic framework has been previously proposed, the original method used conditional CDF of rainfall on discretized duration intervals (Tanaka et al., 2015); the derived flood frequencies by the original method had an uncertainty of discretization of duration. This study newly incorporated a copula approach into the presented flood frequency model. This minor modification enabled less uncertain estimation of flood frequencies. The revised manuscript will try to add this description as well.

[Comment]

*From the methodological point of view, I am skeptical on the adoption of RFFM model. Which is the added value of this procedure compared to the FFA applied on runoff simulations? In my opinion a continuous rainfall-runoff modelling can provide more useful and complete information (indeed it would give also the entire design hydrograph).*

[Response]

Thank you for your comment. The main opinion of this study was that although many previous studies assessed flood risk using frequencies of an upstream boundary discharge based on frequency analysis of observed historical discharge data, it is not appropriate to use this even if discharge data is sufficient because characteristics of river flow during extreme events are largely different from past ones due to upstream inundation, as demonstrated in this study. In this sense, this study accepts that continuous rainfall-runoff modeling approach is also useful for extrapolating frequencies of extreme floods. In the revised manuscript, we will state the motivation and the argument of this study mentioned above much more clearly.

On the other hand, from a methodological viewpoint, we think that the rainfall-based flood frequency model used in this study is new because it theoretically derives the CDF of flood peak discharge, which does not require any Monte Carlo simulation. The revised manuscript will also state this more clearly.

[Comment]

*page 1 line 22: Not only with AMF in general with extreme value sampling procedures.*
Thank you for your comment. This particular approach is major in river basins in Japan.

[Response]

As you pointed out, design flood is decided with some types of extreme values. The revised manuscript will modify this sentence as follows:

"Design flood determines the size of flood control structures such as dams and/or river dikes and is estimated from extreme values of discharge such as annual maximum flood peak discharge (AMF) with 100-year return period."

[Comment]

*page 2 line 6. The added value in considering rainfall data is mainly because these data are largely available compared to runoff data.*

[Response]

Thank you for your comment. As you pointed out, it is true that the main reason of the use of rainfall-based approaches is that rainfall data is much more available than runoff

data. We intentionally stated the change of river basin conditions for highlighting the purpose of this study, but we will revise the manuscript to firstly raise the reason you pointed out followed by the change of river basin conditions.

[Comment]

*page line 17. Literature is very large about this topic, there are many other options not mentioned in the literature review.*

[Response]

Thank you for your comment. As you pointed out, the literature review of this study was limited to initiation of rainfall-based flood frequency model based on the Poisson process, its expansion to areal rainfall and IDF-based approaches. In a revised manuscript, we will also include more reviews of continuous rainfall-runoff modeling approaches.

[Comment]

*page 6 line 9. Poisson instead of Poison*

[Response]

Thank you for your comment. As you pointed out, the revised manuscript will use Poisson instead of Poison.

[Comment]

*page 7 line 15. Nelsen not Nelson (this typo is present also in other part of the text)*

[Response]

Thank you for your comment. As you pointed out, the revised manuscript will use Nelsen instead of Nelson.

[Comment]

*page 7 line 15. It is not clear why the Normal copula was adopted.*

[Response]

Thank you for your comment. In this study, one-parameter families of copula (normal, Gumbel, Frank and Clayton copulas) were fitted to marginal CDF of rainfall and duration ($F_R(r)$ and $F_D(d)$), and the normal copula was selected because the fitted normal copula had the largest AIC. The revised manuscript will describe more detail such as candidates of copula functions and/or fitted parameters.

[Comment]

*In the Introduction authors say that they will compare RFFM approach to observed data, however at the end they use simulated data. I would be consistent and already in the introduction I would say exactly what will be compared at the end.*

[Response]

Thank you for your comment. Although we were careful not to say that discharge data is observed one, page 3 line 7 mistakenly stated observed discharge data. The revised manuscript will more clearly state that the simulated discharge data was used for discharge data-based flood frequency analysis.

---

## Referee Comment (RC2) · R. Uijlenhoet (Referee) · 16 Aug 2016

R. Uijlenhoet (Referee)

remko.uijlenhoet@wur.nl

The paper deals with frequency analysis of extreme floods in managed river basins, where flood control measures have been implemented and where estimates of design floods after implementation of such measures are required. This is a subject of high societal relevance and with appreciable technical and scientific challenges within the broader area of water management. As such, it fits the scope of HESS.

The authors present an application of a largely previously developed (and published) method to a highly developed river basin in Japan. They show that the approach of design flood estimation via a stochastic rainfall generator (calibrated on rainfall time series) based on the Poisson process and hydrologic / hydraulic models is better able to cope with the effects of upstream inundation and/or dams than the classical flood

frequency analysis approach based on historic discharge time series.

This paper adds another case study to an already large body of literature on this topic, which the authors largely fail to acknowledge, not only in the introductory section of their manuscript, but also -and more importantly- in the discussion section. Where does this work fit into the broader scientific context? In what way is it better or worse than previously published work in this field? How can other researchers and practitioners, under different circumstances (in other climates and countries), learn from the presented results? In the current version of their manuscript, the authors fail to tackle these issues. Therefore, the paper largely reads as a report describing a case study, rather than as a scientific paper.

Therefore, I am inclined to recommend rejection of the manuscript in its current form. I have attached an annotated version of the paper to help the authors improve their work. In thoroughly revised form, this work could be resubmitted to HESS or another scientific journal in the area of water management, as far as I am concerned.

Please also note the supplement to this comment:
http://www.hydrol-earth-syst-sci-discuss.net/hess-2016-225/hess-2016-225-RC2-supplement.pdf

———————————————————

[Figure]

**Supplement:**

[revised manuscript text omitted]